# Up/Down-Scaling Photoluminescent Micromarks Written in Diamond by Ultrashort Laser Pulses: Optical Photoluminescent and Structural Raman Imaging

**DOI:** 10.3390/mi13111883

**Published:** 2022-11-01

**Authors:** Pavel Danilov, Evgeny Kuzmin, Elena Rimskaya, Jiajun Chen, Roman Khmelnitskii, Alexey Kirichenko, Nikolay Rodionov, Sergey Kudryashov

**Affiliations:** 1Lebedev Physical Institute, 119991 Moscow, Russia; 2Institution Project Center ITER, 123182 Moscow, Russia

**Keywords:** natural diamond, femtosecond laser, bulk inscription, photoluminescent micromarks, size scaling, 3D scanning Raman and photoluminescence microspectroscopy

## Abstract

Elongated photoluminescent micromarks were inscribed inside a IaAB-type natural diamond in laser filamentation regime by multiple 515 nm, 0.3 ps laser pulses tightly focused by a 0.25 NA micro-objective. The micromark length, diameter and photoluminescence contrast scaled as a function of laser pulse energy and exposure, coming to a saturation. Our Raman/photoluminescence confocal microscopy studies indicate no structural diamond damage in the micromarks, shown as the absent Raman intensity variation versus laser energy and exposition along the distance from the surface to the deep mark edge. In contrast, sTable 3*NV* (*N*3)-centers demonstrate the pronounced increase (up to 40%) in their 415 nm zero-phonon line photoluminescence yield within the micromarks, and an even higher—ten-fold—increase in NV^0^-center photoluminescence yield. Photogeneration of carbon Frenkel “interstitial–vacancy” (I–V) pairs and partial photolytic dissociation of the predominating 2*N* (*A*)-centers were suggested to explain the enhanced appearance of 3*NV*- and *NV*-centers, apparently via vacancy aggregation with the resulting *N* (C)-centers or, consequently, with 2*N*- and *N*-centers.

## 1. Introduction

Ultrashort-pulse laser inscription of photoluminescent micromarks in bulk diamonds recently emerged as a new promising technology for their non-invasive photonic encoding, protecting the original trademarks and the jewelry value of the item [1,2,3]. Generally, it is based on ultrashort-pulse laser structural modification of the present nitrogen impurity centers in a diamond, e.g., usually via expected laser-induced vacancy generation and aggregation with atomic-like nitrogen *C* (*N*)-centers in high-purity electronic-grade synthetic diamonds [4,5,6,7,8,9]. Meanwhile, in natural diamonds with high concentration of nitrogen impurities in different structural forms (*N*, 2*N*, *NV*, 2*NV*, 3*NV*, 4*NV*, 4*N*2*V*, etc. [10]) principal atomistic paths of such ultrashort-pulse laser-induced structural reconfiguration of the present centers are still under active investigation [11,12,13,14]. Moreover, structural damage of host carbon lattice in the laser-induced vacancy-mediated mechanism of structural reconfiguration has not been explored yet [4,5,6,7,8], thus hindering potential applications in diamond photonics.

In this study, we inscribed by ultrashort laser pulses at variable laser pulse energy and exposure an array of photoluminescent micromarks in a bulk natural diamond with a high content of highly aggregated nitrogen impurity centers, and characterized the micromarks regarding lattice damage by 3D scanning confocal Raman microspectroscopy and regarding structural modification of laser-modified nitrogen impurity centers by 3D scanning confocal photoluminescence microspectroscopy.

## 2. Materials and Methods

In this work, we used a workstation for 3D micro- and nanostructuring (Figure 1a) based on a Satsuma femtosecond Yb-doped fiber laser system (Amplitude Systemes), operating at the fundamental wavelength λ = 1030 nm and second harmonic (SH) wavelength λ = 515 nm (TEM00-mode), pulse duration τ ≈ 0.3 ps, variable pulse energy E = 0.01 −10 μJ and repetition rate f = 0–500 kHz. The SH laser pulses were focused by a 0.25 NA micro-objective lens into the spot size w_0_ = 1.8 ± 0.1 μm (1/e-intensity radius) at the depth ∼300 µm inside the diamond cube. Due to the high refractive index value of diamond, n ≈ 2.4, the corresponding spherical aberration was low as for the effective NA = 0.25/2.4 = 0.1. The sample was mounted onto a PC-driven three-dimensional motorized translation stage (Standa) and exposed to a series of bulk micropatterns (micromarks) as a function of peak laser power P = E/τ ≈ 0.8–2.1 MW (peak fluence ≈ 2–5.5 J/cm^2^, peak intensity ≈ 7–18 TW/cm^2^) and exposure time T ≈ 10–240 s (Figure 1b–e) at a 100 kHz repetition rate (pulse exposure N ≈ (1 − 24) × 10^6^ pulses).

The experimental sample was a colorless natural high-nitrogen IaAB-type diamond crystal cube (4 × 4 × 4 mm^3^) (Figure 2a), possessing the average impurity concentrations: [*A* (2*N*)]-centers ~ 230 ppm, [*B*1 (4*NV*)]-centers ~ 50 ppm, some [B2]-centers (platelets), according to FT-IR measurements (Figure 2b), using a Vertex V-70 spectrometer (Bruker, Bruker, MA, USA).

The photoluminescencent (PL) 3D characterization of the inscribed area (Figure 1b,d) was performed by 3D scanning confocal Raman/PL microspectroscopy (inVia InSpect, Renishaw, Great Britain) at the 405 nm laser excitation wavelength and probing depths of 0–400 μm at room temperature (25 °C). PL spectra of the unexposed diamond exhibited *N*3 (3*NV*)-center band (zero-phonon line, ZPL, at 415 nm [10]) and 1332 cm^−1^ Raman band of the triply degenerate optical phonon in diamond (428 nm), while the inscribed micromarks also demonstrated a broad pronounced band of H4 (4*N*2*V*), H3 (2*NV*)-centers in the range of 480–550 nm and NV^0^ band with its ZPL at 575 nm [10] (Figure 2c). Here, the normalized spectrum (Figure 2d) exhibited the laser-induced increase in the PL yield in the range of 500–750 nm.

## 3. Results and Discussion

### 3.1. Power- and Exposure-Dependent Characteristics of Micromarks Visualized by Raman and Photoluminescence Microspectroscopy

The inscribed micromark array was visualized by the PL/Raman microspectroscopy as a function of depth into the diamond. Below, in Figure 3a,d,g, the 3D images indicate the distinct (green–red PL signal, Figure 3a–c), negligible (428 nm Raman signal, Figure 3d–f) and minor (*N*3 ZPL PL signal, Figure 3g–i) variations in the spectra versus the depth in the diamond at the different peak laser powers (Figure 3b,e,h) and exposures (Figure 3c,f,i). These tendencies are then quantitatively overviewed in Figure 4.

Comparing to the Raman (428 nm) and *N*3 ZPL (415 nm) PL signals (Figure 3d–i), respectively, the acquired 3D image and the depth profiles of the much more intense, normalized green–red PL intensity variation (Figure 3a–c) allow measurement of the micromark length, diameter and PL contrast (normalized intensity) as a function of peak laser power and exposure (Figure 4a–c). Specifically, once the peak pulse laser power *P* exceeds the critical power for self-focusing, *P*_cr_, in the natural diamond at the 515 nm laser wavelength (*P*_cr_ ≈ 0.5 MW [15,16]), the extended photoluminescent micromarks could be observed in the diamond bulk in front of the geometrical focal region (Figure 4a). This filamentary inscription regime (*P* ≥ *P*_cr_) enables up- and down-scaling of the micromark lengths according to the square root dependence [17], as well as of micromark diameter and PL contrast (Figure 4a–c). Moreover, the well-known intensity clamping effect in ultrashort-pulse laser filaments provides the corresponding homogeneous energy deposition [17] and the following structural modification in the diamond. The fast exposure-dependent saturation of the micromark length and PL contrast (normalized intensity) (Figure 4b) supports the homogeneous modification in the filamentary tracks in the diamond. Likewise, the micromark diameter remains nearly constant as a function of peak power and exposure due to the same clamping effect, apparently, in the single-filament regime at *P* ~ *P*_cr_ (Figure 4c).

### 3.2. Power- and Exposure-Dependent Structural Modification of Color Centers in Micromarks

The filamentary propagation of the 515 nm, 0.3 ps laser pulses in the diamond results in its local structural modification within the filamentary tracks, emerging as the spectrally contrasting photoluminescent micromarks with the modified initial nitrogen-based color centers *A* (2*N*), *B*1 (4*NV*) and *B*2 (platelets) (Figure 1, Figure 2 and Figure 3). Importantly, such laser-induced modification of the nitrogen impurity centers does not damage the carbon crystalline lattice, as can be seen in the nearly constant 1332 cm^−1^ optical phonon Raman peak intensity as a function of peak laser power and exposure (Figure 3d–f and Figure 4d,e). It means the micromarks appear blind in the Raman spectra due to the rather delicate laser-induced structural modification, not introducing a considerable permanent amount of transient carbon vacancies (*V*) and interstitials (*I*) in the micromarks [18], similar to our other similar studies in natural diamonds [13,14].

Moreover, some minor local increase in N3-center PL yield was identified in the micromarks versus peak laser power and exposure (Figure 4d–e) above the threshold peak power of 1.4 MW and exposure of 60–120 s. In this above-threshold regime, *N*3-related PL yield increases along with the simultaneous increase in the green–red PL intensity (Figure 1c,d and Figure 4d,e). Accounting for the relatively high stability of *N*3-centers, the initial nitrogen impurity center abundance (*A*, *B*1 and *B*2) and the resulting modified PL spectra in the micromarks, the following scheme of atomistic laser-induced reactions could be suggested in this case:


**
*Low laser pulse power*
**

$P<P_{th}$

**
*(<1.4 MW)*
**

(1)
A(N−N)+V→H3(N−V−N)


B1(3N−V−N)+V→H4(3N−V−V−N)


H3(N−V−N)+V→NV+NV


H4(3N−V−V−N)+V→NV+…+NV


B1(3N−V−N)+V→N3(2N−V−N)+NV




**
*High laser pulse power*
**

$P>P_{th}$

**
*(>1.4 MW)*
**

(2)
$$H4\left(3N-V-V-N\right)\overset{h\omega ,T}{\to }N3\left(2N-V-N\right)+NV$$


$$A\left(N-N\right)\overset{h\omega ,T}{\to }C\left(N\right)+C\left(N\right)$$


NV+C→H3 (N−V−N);H3(N−V−N)+C→N3(2N−V−N)



Here, the sub-threshold low-power regime (1) apparently could be related to vacancy-driven aggregation (*H*3-, *H*4-centers—green PL fraction in Figure 1c,d) or nitrogen detachment from *NV*-centers (yellow–red PL fraction in Figure 1c,d) and, to a lesser extent, *N*3-centers (Figure 3 and Figure 4). In contrast, the above-threshold high-power regime (2) could involve, mostly, photodecomposition of the initial *A*- and *H*4-centers, while the product nitrogen atoms could join *NV*- and *H*3-centers to result in the rather stable terminal *N*3-centers, accumulating in the micromarks in this regime (Figure 4d,e). At least, *NV*-based PL signal saturates and even decreases versus laser power in Figure 4d, contrasting to the *N*3-based PL yield.

The basic structural processes observed in this work for the first time during ultrashort-pulse laser exposure of natural diamond go beyond the previously considered elementary reaction *N* + *V* → *NV* in ultrapure synthetic diamonds [4,5,6,7,8,9], reflecting the plethora of different nitrogen impurity centers and the variety of available photo-, pressure- and thermally driven [19] atomistic transformations in natural and synthetic diamonds [10,20,21,22]. Steady-state post-processing thermal annealing of the fs laser inscribed micro-regions could favorably enhance their PL contrast, similarly to *NV*-center PL yield in ultrapure synthetic diamond [4], but is not relevant for our micromarking technology [1,2,3]. Moreover, more studies are obviously required to distinguish between laser-induced non-thermal (I–V generation), transient stress-driven and thermally driven structural transformations of nitrogen color centers in diamonds [10,19,20,21,22,23,24].

## 4. Conclusions

Photoluminescent micromarks were inscribed in bulk natural diamond by 515 nm, 0.3 ps laser pulses in the filamentary regime, enabling up- or down-scaling of micromark length, diameter and photoluminescence contrast (normalized intensity) as a function of peak laser power and, to lesser extent, laser exposure. The micromarks appear to spectrally contrast in PL spectra at the 405 nm laser excitation due to ultrashort-pulse laser-induced structural modification in the diamond, occurring in two power-dependent characteristic regimes. The first, sub-threshold regime (*P* < 1.4 MW) is characterized by the strong enhancement of NV-center emission, apparently via laser-generated transient vacancy-driven aggregation of nitrogen impurity centers or nitrogen detachment. The second, above-threshold regime (*P* > 1.4 MW) apparently indicates photodecomposition of initial A- and H4-centers, with the product nitrogen atoms joining intermediate NV- and H3-centers to result in the terminal N3-centers. The identified ultrashort-pulse laser exposure regimes and anticipated underlying laser-induced structural atomistic transformations in the natural diamond open new technological opportunities in diamond photonics and micromachining.

## Figures and Tables

**Figure 1 micromachines-13-01883-f001:**
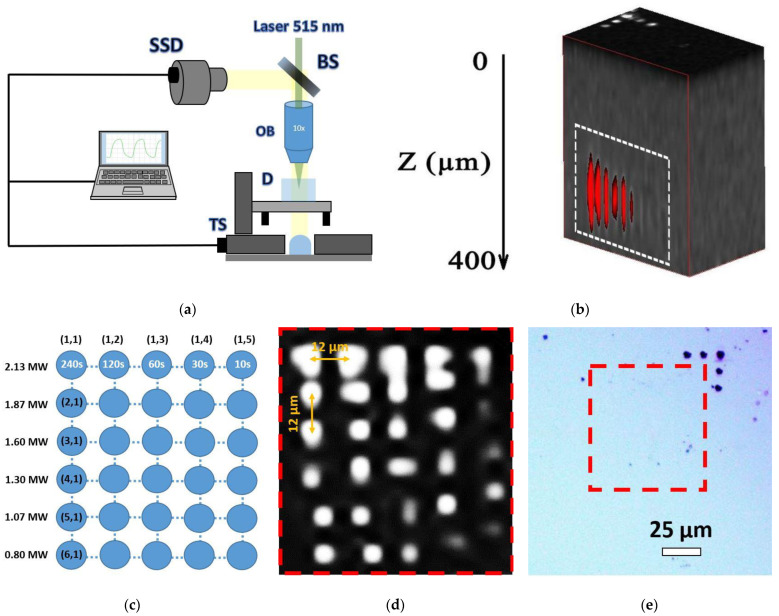
(**a**) Scheme of workstation for direct writing of micromarks inside the natural diamond by 515 nm, 0.3 ps laser pulses: BS—beam splitter, OB—0.25 NA micro-objective lens, D—diamond sample, CCD—CCD camera, TS—3D translation stage. (**b**) The 3D reconstruction of the inscribed region with micromarks (white area, red stripes) visualized by 3D confocal Raman microscope at 405 nm laser pump wavelength. The depth (**left** axis) varies in the range of 0–400 μm. (**c**) Schematic of micromark array with corresponding inscription parameters (peak laser power P ≈ 0.8–2.1 MW, peak fluence—2–5.5 J/cm^2^, peak intensity—7–18 TW/cm^2^, exposure time T ≈ 10–240 s, (1 − 24) × 10^6^ pulses). (**d**) Cross-sectional PL image of the micromark array at 550 nm wavelength. (**e**) Optical microscopy image of the diamond surface above the micromark array with the few damage spots made for fast navigation in the sample (see also on the **top** face in (**b**)).

**Figure 2 micromachines-13-01883-f002:**
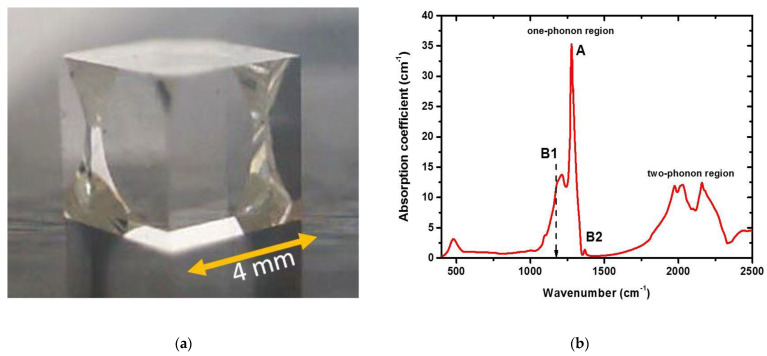
(**a**) Image of the natural IaAB-type diamond cubic sample. (**b**) FT-IR spectrum of the initial diamond with its spectral assignment after [10]. (**c**) PL spectra of the reference diamond region (blue curve, **bottom** one) and its micromark (green curve, **top** one) inscribed at 2.1 MW (fluence—5.5 J/cm^2^, intensity—18 TW/cm^2^) and 240 s (24 × 10^6^ pulses) at the 280 μm depth. Spectral assignment after [10]. (**d**) Spectrum of normalized PL intensity, obtained by dividing the second spectrum in (**c**) by the first one (micromark/reference). Spectral assignment after [10].

**Figure 3 micromachines-13-01883-f003:**
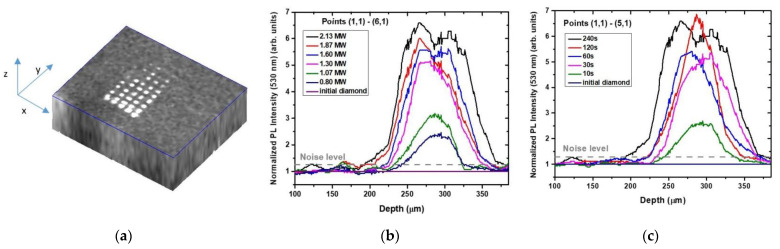
(**a**) The 3D PL map of the inscribed region with micromarks at the 550 nm wavelength. (**b**) Depth profiles of the micromarks inscribed at different peak powers and 240 s exposure time (24 × 10^6^ pulses). (**c**) Depth profiles of the micromarks inscribed at different exposure times and 2.1 MW peak power (fluence—5.5 J/cm^2^, intensity—18 TW/cm^2^). (**d**) The 3D Raman map of the inscribed region with micromarks at the 428 nm wavelength. (**e**) Depth profiles of the micromarks inscribed at different peak powers and 240 s exposure time. (**f**) Depth profiles of the micromarks inscribed at different exposure times and 2.1 MW peak power. (**g**) The 3D map of the inscribed region with micromarks at the 415 nm wavelength. (**h**) Depth profiles of the micromarks inscribed at different peak powers and 240 s exposure time. (**i**) Depth profiles of the micromarks inscribed at different exposure times and 2.1 MW peak power.

**Figure 4 micromachines-13-01883-f004:**
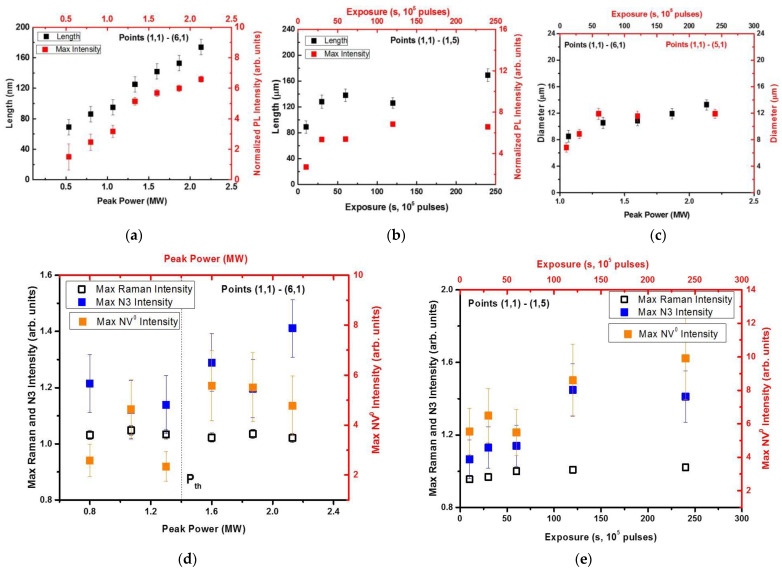
(**a**) Micromark length (dark symbols, **left** axis) and normalized maximal intensity of the peak at 550 nm wavelength (red symbols, **right** axis) versus peak power at 240 s exposure time (24 × 10^6^ pulses). (**b**) Micromark length (dark symbols, **left** axis) and normalized maximal intensity of the peak at 550 nm wavelength (red symbols, **right** axis) versus exposure time at 2.1 MW peak power (fluence—5.5 J/cm^2^, intensity—18 TW/cm^2^). (**c**) Micromark diameter (spot size) versus peak power at 240 s exposure time (dark symbols, **left** and **bottom** axes) and exposure time at 2.1 MW peak power (red symbols, **right** and **top** axes). (**d**) Micromark normalized maximal intensity at 428 nm (Raman signal, dark open squares) and 415 nm (N3 PL signal, blue squares) wavelength (**left** and **bottom** axes), and at 575 nm (NV^0^ PL signal, **right** and **top** axes) wavelength versus peak power at 240 s exposure time. (**e**) Micromark normalized maximal intensity at 428 nm (Raman signal, dark open squares) and 415 nm (N3 PL signal, blue squares) wavelength (**left** and **bottom** axes), and at 575 nm (NV^0^ PL signal, **right** and **top** axes) wavelength versus exposure time at 2.1 MW peak power.

## Data Availability

Additional data could be provided by the authors upon a special request.

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
