# Peer review of "Up/Down-Scaling Photoluminescent Micromarks Written in Diamond by Ultrashort Laser Pulses: Optical Photoluminescent and Structural Raman Imaging"

_micromachines, 2022, doi:10.3390/mi13111883_

Round 1

Reviewer 1 Report

Dear Authors,

congratulation to your work and to your results. I like the way you do your research.

The introduction and the conclusion encourage for reading your article.

Those summaries are supported by your results. Unluckily the presentation of the results was somehow disappointing for me. You show a lot of pictures, you give detailed information under the pictures. for me it was too much, too many details. In the end it was hard for me to identify your conclusion.

Do you see a chance to draw the eye of the reader more obviously to your interpretation without guiding it too strongly, so there might be room for her / his own ideas? in the end there was too little interpretion of what could be seen in the grafical displays of your experimental results. eg. Figure 3 if you compare f to i or e to h. in Figure 3 you combine very different  information and multiple of graphs. What do think about splitting them up into packs which belong together in your opinion and help you draw your conclusion?

may you please check the last sentence of your abstract, especially "... via vacancy aggregation with the resulting N centers and, consequently, with 2N and N centers."

line 89: microspectroscopy is missing the t.

I appreciate your work and the your results. Thank you very much.

Reviewer 2 Report

the results of luminescence and Raman scattering from laser inscribed marks in diamond are discussed. really, just discussed with model of interactions presented. usual analysis includes thermal annealing and extraction of activation energies between defect changes and clustering. this would be very useful to add. it should help with model. 

pulse power is used for definition of the inscription conditions. would be useful to add fluence and intensity. spherical aberration is huge and affects focusing. how inscription is changed when depth of focus or NA of lens is different. current results are experiment specific. 

Round 2

Reviewer 1 Report

Dear Authors, thank you for your effords you put additionally to your report. May you please hav a look at fig. 4: some of the graphics (a-c) look different to the other two (d and e). your figures are still very big, that is the explanation to individual pictures is far away and not easy to detect. I appreciate very much that you give so many detailed results of your research. I personally would be very glad to get some kind of overview as well. Taking into account the size of your figures and the space on one page in the magazine, information of one figure is split up to different pages (fig 2, 3 and 4).

Reviewer 2 Report

remarks answered